# Characterization of Austenitic Stainless Steels with Regard to Environmentally Assisted Fatigue in Simulated Light Water Reactor Conditions

**Matthias Bruchhausen [1],***, **Gintautas Dundulis [2]** , **Alec McLennan [3]**, **Sergio Arrieta [4],†**, **Tim Austin [1]**, **Román Cicero [5]**, **Walter-John Chitty [6]**, **Luc Doremus [7]**, **Miroslava Ernestova [8]**, **Albertas Grybenas [2]**, **Caitlin Huotilainen [9]**, **Jonathan Mann [10]**, **Kevin Mottershead [3]** , **Radek Novotny [1]**, **Francisco Javier Perosanz [4]**, **Norman Platts [3]**, **Jean-Christophe le Roux [11]**, **Philippe Spätig [12,13]**, **Claudia Torre Celeizábal [5]**, **Marius Twite [10,‡]** and **Marc Vankeerberghen [14]**

[1] European Commission, Joint Research Centre, Westerduinweg 3, 1755 LE Petten, The Netherlands; Timothy.AUSTIN@ec.europa.eu (T.A.); radek.novotny@ec.europa.eu (R.N.)

[2] Lithuanian Energy Institute, Breslaujos 3, LT-44403 Kaunas, Lithuania; gintautas.dundulis@lei.lt (G.D.); albertas.grybenas@lei.lt (A.G.)

[3] Jacobs, Faraday Street, Birchwood Park, Warrington WA3 6GA, UK; alec.mclennan@jacobs.com (A.M.); Kevin.Mottershead@jacobs.com (K.M.); norman.platts2@jacobs.com (N.P.)

[4] Structural Materials Division, Technology Department, Centro de Investigaciones Energéticas, Medioambientales y Tecnológicas (CIEMAT), Avenida Complutense 40, 28040 Madrid, Spain; sergio.arrieta@unican.es (S.A.); franciscojavier.perosanz@ciemat.es (F.J.P.)

[5] Inesco Ingenieros, CDTUC, Fase B, Av. Los Castros 44, 39005 Santander, Spain; ciceror@inescoingenieros.com (R.C.); claudia.torre@inescoingenieros.com (C.T.C.)

[6] Institut de Radioprotection et de Sûreté Nucléaire (IRSN), PSN-RES/SEREX/LE2M, CEN de Cadarache, 13115 St. Paul lez Durance, France; walter-john.chitty@irsn.fr

[7] Framatome, Technical Centre, 30 Bd de l'industrie, 71200 Le Creusot, France; luc.doremus@framatome.com

[8] UJV REZ, a. s., Hlavni 130, 25068 Husinec, Czech Republic; miroslava.ernestova@ujv.cz

[9] VTT Technical Research Centre of Finland, Nuclear Reactor Materials, Kivimiehentie 3, 02150 Espoo, Finland; caitlin.huotilainen@vtt.fi

[10] Rolls-Royce, Raynesway P.O. Box 2000, Derby DE21 7XX, UK; jonathan.mann2@rolls-royce.com (J.M.); emmarty@yahoo.com (M.T.)

[11] EDF—R&D MMC, Avenue des Renardières—Ecuelles, 77250 Moret-Loing et Orvanne, France; jean-christophe.le-roux@edf.fr

[12] Laboratory for Nuclear Materials, Paul Scherrer Institute, 5232 Villigen-PSI, Switzerland; philippe.spatig@psi.ch

[13] Laboratory for Reactor Physics and Systems Behaviour, Ecole Polytechnique Fédérale de Lausanne, 1015 Lausanne, Switzerland

[14] SCK CEN, Boeretang 200, 2400 Mol, Belgium; mvankeer@sckcen.be

* Correspondence: matthias.bruchhausen@ec.europa.eu

† Current address: Laboratory of Materials Science and Engineering (LADICIM), University of Cantabria, E.T.S. de Ingenieros de Caminos, Canales y Puertos, Av. Los Castros 44, Santander, 39005 Cantabria, Spain.

‡ Current address: Emmarty Ltd., Weston-super-Mare BS23 4UL, UK.

**Abstract:** A substantial amount of research effort has been applied to the field of environmentally assisted fatigue (EAF) due to the requirement to account for the EAF behaviour of metals for existing and new build nuclear power plants. We present the results of the European project INcreasing Safety in NPPs by Covering Gaps in Environmental Fatigue Assessment (INCEFA-PLUS), during which the sensitivities of strain range, environment, surface roughness, mean strain and hold times, as well as their interactions on the fatigue life of austenitic steels has been characterized. The project included a test campaign, during which more than 250 fatigue tests were performed. The tests did not reveal a significant effect of mean strain or hold time on fatigue life. An empirical model describing the fatigue life as a function of strain rate, environment and surface roughness is developed. There is evidence for statistically significant interaction effects between surface roughness and the environment, as well as between surface roughness and strain range. However, their impact on fatigue life is so small that they are not practically relevant and can in most cases be neglected. Reducing the environmental impact on fatigue life by modifying the temperature or strain rate leads to an increase of the fatigue

life in agreement with predictions based on NUREG/CR-6909. A limited sub-programme on the sensitivity of hold times at elevated temperature at zero force conditions and at elevated temperature did not show the beneficial effect on fatigue life found in another study.

**Keywords:** environmentally assisted fatigue (EAF); austenitic stainless steel; nuclear power plant (NPP); light water reactor (LWR); surface roughness

## 1. Introduction

According to the most recent report of the Intergovernmental Panel on Climate Change (IPCC), "Nuclear energy is a mature low-GHG [green house gas] emission source of baseload power, but its share of global electricity generation has been declining (since 1993). Nuclear energy could make an increasing contribution to low-carbon energy supply, but a variety of barriers and risks exist" [1]. Hence, long-term operation (LTO) of the current fleet of nuclear power plants (NPPs) can make an important contribution to controlling GHG emissions, especially in the short term. However, this requires proper understanding of the relevant damage mechanisms in NPPs.

Environmentally assisted fatigue (EAF) is one of these damage mechanisms; test programmes in Japan, the U.S. and later in Europe have shown that the water environment in NPPs reduces the fatigue life $N_f$ significantly. Nevertheless, EAF was not explicitly taken into account during the construction of the currently operating fleet of NPPs [2,3]. The most recent guidance for EAF assessment is the U.S. regulation NUREG/CR-6909, Rev. 1 [4], in its final version from May 2018, which is based on an extensive collection mainly of Japanese and U.S. data. In that document, the effect of the environment on $N_f$ is described by an environmental factor $F_{en}$:

$$F_{en} = \frac{N_{f,air,RT}}{N_{f,LWR}} \tag{1}$$

where $N_{f,air,RT}$ is the fatigue life in air at room temperature and $N_{f,LWR}$ the fatigue life in the environment at operating conditions.

However, the low cycle fatigue lives predicted by CR-6909 do not reflect current pressurized water reactor (PWR) plant experience where no failures attributed to environmental fatigue have been observed so far where the loading conditions were known [2]. Furthermore, studies on laboratory specimens found experimental fatigue lives to be longer than predictions based on CR-6909 [5,6]. This indicates that the guidance provided by CR-6909 includes significant conservatism, which could potentially be reduced without loss of operational plant safety. Accordingly, EAF has received much attention in the last few years [5–15].

Work by Chopra et al. presented a recent review with a focus on ASME Code section III [3] where the $F_{en}$ for austenitic stainless steels is described as:

$$F_{en} = \exp(-T^* \dot{\varepsilon}^* O^*) \tag{2}$$

$T^*$, $\dot{\varepsilon}^*$ and $O^*$ are functions of the environmental temperature, the positive strain rate and the dissolved oxygen content. Other parameters like surface finish and complex waveforms are not explicitly taken into account, but taken into account through constant subfactors [4].

However, some authors have observed cases where the combined effect of surface finish and the environment is less damaging than might be expected when considering both effects independently [10,11,16]. These findings suggest there might be interaction effects between the surface finish and the environment.

Similarly, a number of studies investigated the influence of the waveform and especially mean strain [8] and hold time periods on environmental fatigue [6,13]. While mean strain did not have a major effect on fatigue life, it turned out that at least under certain

conditions, introducing hold times at some cycles during the fatigue life can extend the fatigue life of austenitic steels in the PWR environment.

The project INCEFA-PLUS (INcreasing Safety in NPPs by Covering Gaps in Environmental Fatigue Assessment) [17] was started in 2015 under the umbrella of the European Horizon 2020 programme to characterize some of this conservatism. It includes a major test programme with more than 250 (mostly strain controlled) fatigue tests in air and a simulated light water reactor (LWR) environment carried out in 11 European laboratories. While most of the tests were carried out according to a single test matrix that was optimized by the design of experiments method, some specific aspects were addressed in separate sub-programmes.

This work describes the test programme in detail and analyses the data from the main programme in which the effects of five test parameters, as well as their two-factor interactions are considered. The most relevant factors and interactions are identified. Two sub-programmes respectively address hold time effects and conditions under which less environmental impact is expected (i.e., smaller $F_{en}$).

The implications for actual plant assessment were discussed elsewhere [18]. The analyses presented there were based on an earlier data evaluation similar to the one presented here, but based on a slightly smaller database. The conclusions for fatigue assessment in plants are not affected by the small difference in the underlying database.

## 2. Materials

The large majority (86%) of the tests were carried out on a single batch (XY182 sheet 23201) of 304L stainless steel produced by Creusot Loire Industries. The remaining tests were carried out on a single batch of 321 (8%) and different batches of 304, 304L and 316L. The chemical composition of the different steels is listed in Table 1. All materials were annealed at temperatures between 1050 °C and 1100 °C. The annealing time was 5 h for the 321 material and between 0.4 and 2 h for the other steels.

**Table 1.** Chemical composition of the different steels (wt.%). The common material (see column "Comment") was used in the majority of the tests; the other materials were only used by the indicated organizations.

| Material | Al | B | C | Co | Cr | Cu | Fe | Mn | Mo | N |
|---|---|---|---|---|---|---|---|---|---|---|
| 304L | | | 0.029 | | 18.00 | 0.02 | bal. | 1.86 | 0.04 | 0.056 |
| 304L | 0.029 | 0.0005 | 0.026 | 0.016 | 18.626 | 0.046 | bal. | 1.558 | 0.227 | 0.074 |
| 316L | 0.022 | 0.001 | 0.028 | 0.007 | 17.562 | 0.049 | bal. | 1.779 | 2.393 | 0.062 |
| 304 | | | 0.035 | 0.05 | 18.39 | 0.17 | bal. | 1.83 | 0.2 | 0.079 |
| 321 | 0.109 | | 0.102 | | 18.08 | 0.048 | bal. | 1.446 | 0.023 | |

| Material | Nb | Ni | P | S | Si | Ta | Ti | V | W | Comment |
|---|---|---|---|---|---|---|---|---|---|---|
| 304L | | 10.00 | 0.029 | 0.004 | 0.37 | | | | | Common |
| 304L | 0.003 | 9.737 | 0.0133 | 0.0005 | 0.527 | 0.01 | | | | IRSN |
| 316L | 0.002 | 11.947 | 0.0121 | 0.0084 | 0.642 | 0.01 | | | | IRSN |
| 304 | | 8.07 | 0.031 | 0.001 | 0.32 | | | | 0.05 | Jacobs |
| 321 | | 9.79 | 0.023 | | 0.52 | | 0.61 | 0.013 | | UJV |

## 3. Test Programme

The test programme consisted of a main programme and several sub-programmes dedicated to specific questions arising during the project.

The main programme initially aimed at studying the sensitivities of fatigue life $N_f$ to the parameters strain range $\varepsilon_r$ (difference between the maximum and minimum strain during the test), mean strain $\varepsilon_m$ (strain level in the middle between the minimum and the maximum strain in a test), hold time $t_h$ (period with constant strain), surface roughness $R_t$ and environment $E$. These parameters were selected based on the interest of the project partners and the EPRI gap report [2]. The main programme was divided into three consecutive phases (see Section 3.1 for details) to be able to refocus the testing once the first trends became apparent in the data. The data from Phase I did not show any effect of mean strain ([19] and Section 3.1.3), and this was dropped in the later phases. The factor mean strain $\varepsilon_m$ was introduced as the easiest means to simulate the constant load applied to NPP

components during steady state operation. However, because of shake down early during the test, the mean strain did not have a significant effect on fatigue life. A sub-programme was started to simulate the constant load via tests with mean stress under strain control. The results of this sub-programme were published separately [20].

It also became apparent that applying hold times during some cycles in this study did not have a major effect on fatigue life. However, significant effects of hold times on fatigue life were reported in a different study [13]. To investigate whether differences in the application of holds led to these differences, a limited sub-programme on hold time tests was started (Section 3.3).

Furthermore, a small test programme with conditions where less environmental effects were expected (i.e., smaller $F_{en}$) than in the main programme was performed (Section 3.2).

In the absence of a dedicated standard for EAF tests in LWR conditions, the tests were performed as much as possible according to ISO 12106:2017, the standard for strain controlled fatigue testing [21] with additional guidance taken from other relevant standards such as ASTM E606 [22], ISO 11782-1 [23], BS7270:2006 [24] and AFNORA03-403 [25]. To reduce the scatter caused by differences in testing practices between the different laboratories, further guidelines were developed that provide more detailed guidance than is normally included in a testing standard [26]. All test data were uploaded in a dedicated materials database operated by the European Commission (MatDB) and have received digital object identifiers (DOIs) to ensure long-term storage and traceability.

As an additional quality assurance measure, each test was validated by a panel of fatigue experts from within the project and rated with regard to the quality of the test and the completeness of the information in the database. The test quality was determined on the basis of data like the cyclic stress amplitudes and hysteresis curves [26]. Where necessary, this information was complemented by microstructural characterization of the specimens [27]. From the strain controlled tests carried out during the project, ninety-four percent received a quality rating of one or two (out of four) and were accepted without restriction for analysis.

Besides these tests on uni-axial specimens, the project included also a sub-programme on membrane specimens. The results of this sub-programme were published separately [28].

### 3.1. Main Programme

### 3.1.1. Test Conditions

The main test programme addressed the influences of the factors strain range $\varepsilon_r$, hold time $t_h$, surface roughness characterized by the total height of the roughness profile $R_t$, mean strain $\varepsilon_m$ and environment on the fatigue life of austenitic stainless steels. Preliminary data analyses yielded no indications of significant effects of mean strain and hold times on $N_f$ and were removed during the later phases of the test programme.

Each of the three test phases was optimized by means of the design of experiments (DOE) method [29]. As usual for experimental campaigns for linear models optimized by DOE, all factors were tested on two levels. In the case of continuous factors (like strain range), the minimum and maximum values in the interval of interest were chosen. Using the extreme values maximizes the sensitivity of the test result on the factor settings (because of the higher leverage of the extreme values compared to intermediate values). The only exception from this rule is the surface roughness: While all smooth specimens have a very similar surface roughness, the grinding process used to obtain the rougher surface finishes yielded a roughness distribution rather than a discreet value (Section 3.1.2). For categorical factors like hold time, the two levels were "without holds" and "with holds".

Table 2 lists the test conditions applied in the main programme. The test conditions were selected to be as plant relevant as possible while keeping test durations realistic (especially for hold times and minimum strain range). The surface roughness is characterized for all specimens by the maximum roughness height $R_t$ and average roughness $R_a$ as specified in ISO 4287:1997 [30]. The smooth surface finishes achieved by polishing

were very reproducible, so not all specimens were measured individually, and generic roughness values were used for most polished specimens. Because of the larger scatter in the surface roughnesses of the rough specimens, $R_a$ and $R_t$ for all ground specimens were measured individually by optical confocal profilometry according to ISO 4288:1997 [31]. In this work, $R_t$ is used rather than $R_a$ because that is the parameter that can be expected to have more impact on crack initiation: a deeper scratch leads to larger stress concentration, which facilitates crack initiation. As both values are strongly correlated (Figure 1), the choice of the surface characteristic is not expected to have a major impact on the analysis.

**Table 2.** Test conditions in the main programme and the sub-programme on low $F_{en}$ testing.

| Parameter | Low Level | Middle Level | High Level | Comment |
|---|---|---|---|---|
| $\varepsilon_r$ (%) | 0.6 | | 1.2 | |
| $\varepsilon_m$ (%) | 0 | | 0.5 | only for Phase I |
| $R_t$ (μm) | 0.76 | ≈20 | >40 | $R_t$ > 40 for Phase II only |
| $t_h$ (h) | 0 | | 72 | 0 or 3 holds of 72 h at mean strain; cycles with holds depend on test conditions |
| $\dot{\varepsilon}$ (%/s) | 0.01 | | 0.1 | rising $\dot{\varepsilon}$ in PWR env., falling $\dot{\varepsilon}$ and air tests may vary; $\dot{\varepsilon}$ = 0.1 %/s in low $F_{en}$ tests only |
| $T$ (°C) | 230 | | 300 | $T$ = 230 °C in low $F_{en}$ tests only |

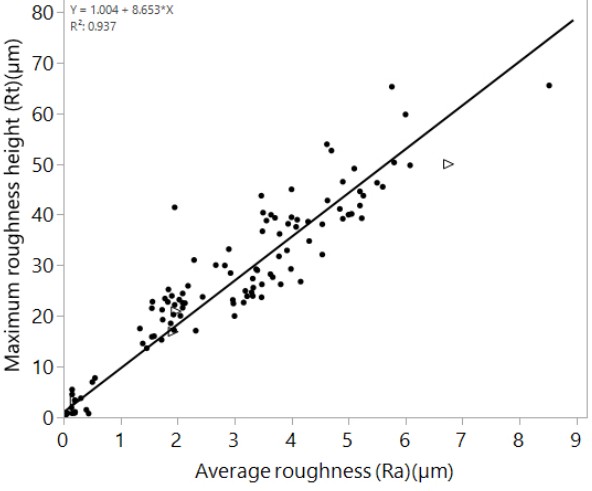

**Figure 1.** Correlation between $R_t$ and $R_a$ for the specimens in the database (throughout this work, the symbol ▷ indicates runout specimens). The ratio between $R_t$ and $R_a$ is 8.7.

The PWR and VVER (a Russian PWR design) chemistries are defined in Table 3. In some cases, slightly different water chemistries were used because of different practices in the national power plants. These differences are not expected to have a significant impact on fatigue life, but are recorded in the central database. All tests with the material 321 (and only these) were performed in the VVER environment.

**Table 3.** Definition of the water chemistry; $\sigma_e$ is the electric conductivity; DH2 and DO are the dissolved hydrogen and oxygen contents.

| Reactor | $T$ °C | $p$ MPa | pH @ 300 °C | Li ppm | B ppm | K ppm | NH₃ ppm | DH2 cc(STP)H₂/kg | DO ppb | $\sigma_e$ @ 25 °C μS/cm |
|---|---|---|---|---|---|---|---|---|---|---|
| PWR | 300 | 15 | 6.95 | 2 | 1000 | | | 25 | <5 | 30 |
| VVER | 300 | 12.5 | 7 | | 1189 | 16.4 | 9.7 | 2 | 22 | 80–110 |

According to CR-6909, the $F_{en}$ for austenitic stainless steels can be formulated as [4]:

$$F_{en} = \exp(-T^* \dot{\varepsilon}^* O^*) \tag{3}$$

where $T^*$, $\dot{\varepsilon}^*$ and $O^*$ are the parameters derived from temperature, strain rate and dissolved oxygen content. For the conditions used in this work (Tables 2 and 3), these are defined as:

$$T^* = (T - 100)/250 \text{ where } T \text{ is in } °C \tag{4}$$

$$\dot{\varepsilon}^* = \ln(\dot{\varepsilon}/7) \text{ where } \dot{\varepsilon} \text{ is in } \%/s \tag{5}$$

$$O^* = 0.29 \tag{6}$$

For the test conditions in the main programme ($T = 300\ °C$, $\dot{\varepsilon} = 0.01\ 1/s$, Tables 2 and 3), Equation (3) yields $F_{en} = 4.57$.

### 3.1.2. Data Overview

Figure 2 shows the 170 tests that were included in the analysis of the data from the main programme [32]. Four of these tests were runouts, i.e., tests that were stopped for other reasons than specimen failure. These tests are considered as right censored data in the analysis. The mean air curve for austenitic steels from NUREG/CR-6909 [4] and the same curve divided by the $F_{en}$ are plotted for reference.

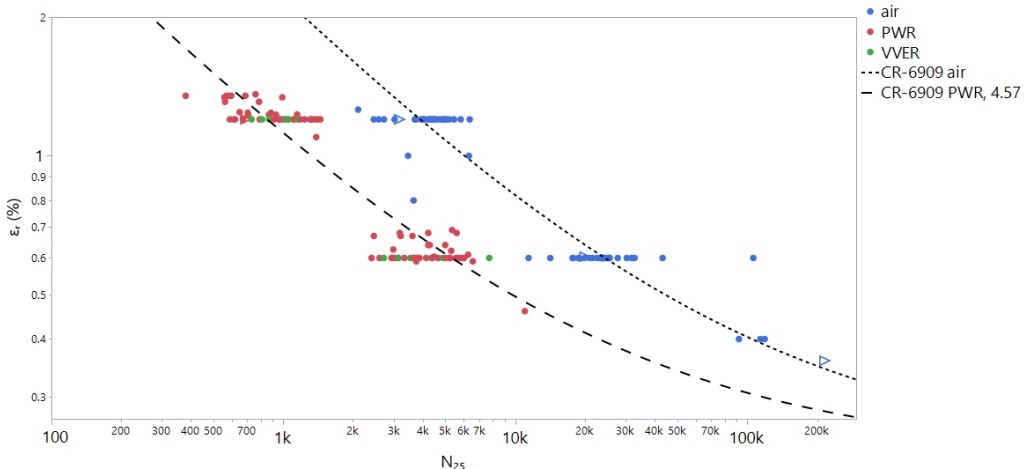

**Figure 2.** Data from the main programme ($F_{en} = 4.57$). The colours refer to the test environment. The ▷ indicate runouts, i.e., tests that were stopped before specimen failure (e.g., because of a technical problem with the test rig).

The definition of fatigue life $N_f$ used in this study is $N_{25}$, i.e., the cycle where a reduction of the maximum cyclic stress of 25% compared to the extrapolated stabilized behaviour occurs. In cases where $N_X$ values other than $N_{25}$ are reported, these were converted to $N_{25}$ by means of Equation (18) in NUREG/CR-6909 [4]:

$$N_{25} = \frac{N_X}{0.947 + 0.00212X} \tag{7}$$

While the majority of the tests in the environment were carried out using solid specimens in autoclaves, some data were acquired on hollow specimens where the water flows through the specimen. For hollow specimens, $N_f$ is generally the cycle where leakage occurs. This is considered a rough equivalent to $N_{25}$ [4].

Each organization used their own specimen type and geometry. For the air tests, the specimen diameters varied between 3.6 and 10.0 mm; the solid specimens for the tests in the environment had diameters between 3.6 and 9.0 mm. The hollow specimens had inner diameters between 9 and 12 mm.

Because of the internal pressure in hollow specimens, the stress state in hollow specimens is different from the membrane stress in solid specimens. It is therefore not obvious that the fatigue lives obtained with both types of specimens can be compared

directly [33–35]. A study carried out within INCEFA-PLUS led to the conclusion that no significant effect on the mean values is expected for the data discussed here [36] (this analysis was done on an earlier (smaller) data set, but has been confirmed with the final dataset). Therefore, no further distinction between the two types of specimens is made here. For hollow specimens, the strains are used directly as measured, and no strain correction as suggested in [35] was applied.

The distribution of the independent variables in the main programme is summarized in Figure 3. The relatively low number of tests with a positive mean strain $\varepsilon_m$ and a positive hold time $t_h$ reflects the fact that these parameters were dropped in Test Phases II and III, respectively (Table 2).

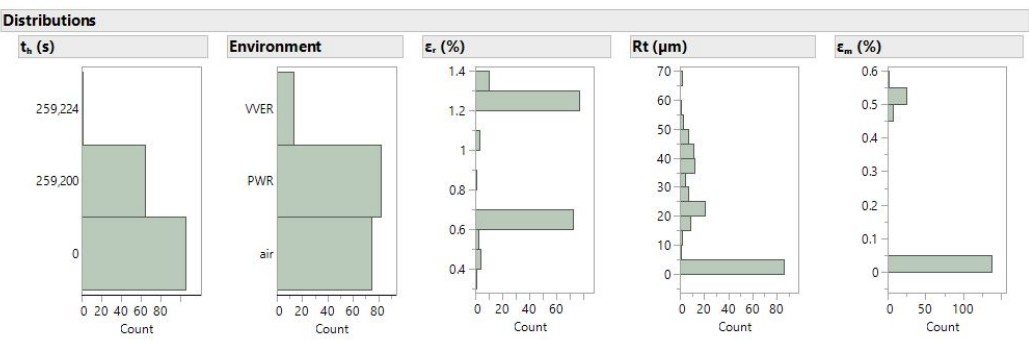

**Figure 3.** Distribution of the factors in the main programme.

### 3.1.3. Data Analysis

Before starting the actual data analysis, it is useful to check for possible correlations. The correlation $r_{i,j}$ between the input parameters $x_i$ and $x_j$ is given by:

$$r_{i,j} = \frac{\sum (x_i - \overline{x}_i)(x_j - \overline{x}_j)}{\sqrt{\sum (x_i - \overline{x}_i)^2} \sqrt{\sum (x_j - \overline{x}_j)^2}} \tag{8}$$

where $\overline{x}$ is the mean of $x$. The correlation $r_{i,j}$ can take values in the interval $[-1;1]$. Values of $|r_{i,j}|$ close to 1 indicate strong (anti-)correlations. If $|r_{i,j}|$ is close to 0, $x_i$ and $x_j$ are not correlated. A strong correlation between $x_i$ and $x_j$ means that tests with high values of $x_i$ also tend to have high values for $x_j$. Similarly, a strong anticorrelation between $x_i$ and $x_j$ means that high values for $x_i$ are often associated with low values for $x_j$. Strong (anti-)correlations between the inputs can easily lead to the wrong conclusions during the evaluation because the associated effects cannot be separated.

The three phases of the main test programme were optimized by the design of experiments method [29], which also minimises the correlation between the factors. However, the available collection of tests varied from the planned test matrix, since some tests were invalid or not carried out as specified. Furthermore, additional data were contributed by some project partners, and some test conditions were modified during the project. These circumstances could have introduced correlations between the independent variables.

Table 4 lists the correlations between the factors in the main programme. The largest (anti-)correlations were found between $\varepsilon_m$ and $R_t$ and between $\varepsilon_r$ and $R_t$. An anticorrelation between $\varepsilon_m$ and $R_t$ was expected since in Phases II and III, no tests with holds were carried out any more, whereas in Phase II, a higher surface roughness $R_t$ was introduced. Therefore, one would expect tests with holds to have on average lower $R_t$ and hence an anticorrelation between $R_t$ and $t_h$. The correlation between $\varepsilon_r$ and $R_t$, however, is unexpected. Most likely, it is a random effect resulting from the grinding process that was used to produce the rough surface finishes and that yielded a distribution of surface roughnesses rather than specific $R_t$ values (Figure 3). These two largest (anti-)correlations were below 0.15 and should not have a major impact on the evaluation.

**Table 4.** Correlations between the factors in the main programme

|  | $\varepsilon_r$ | $\varepsilon_m$ | $R_t$ | $t_h$ | $E$ |
|---|---|---|---|---|---|
| $\varepsilon_r$ | 1.0000 | −0.0183 | 0.1443 | 0.0252 | 0.0492 |
| $\varepsilon_m$ | −0.0183 | 1.0000 | −0.1402 | −0.0168 | −0.0089 |
| $R_t$ | 0.1443 | −0.1402 | 1.0000 | 0.0952 | −0.0094 |
| $t_h$ | 0.0252 | −0.0168 | 0.0952 | 1.0000 | 0.0409 |
| $E$ | 0.0492 | −0.0089 | −0.0094 | 0.0409 | 1.0000 |

The actual data analysis was based on a second degree factorial model, i.e., a model including the main effects and all second order interactions:

$$\ln(N_f) = \sum_i \alpha_i x_i + \sum_{i<j} \alpha_{ij} x_i x_j + I \tag{9}$$

The $x_i$ are the different factors (such as $R_t$). The parameters $\alpha_i$ and $\alpha_{ij}$ are the model parameters for the main effects and the two factor interactions, and $I$ is the intercept. For every test, an equation like Equation (9) is formulated. The best model is the model for which the parameters $\alpha_i$, $\alpha_{ij}$ and $I$ best describe the experimental data. A lognormal distribution for $N_f$ is assumed as recommended in ISO 12107 [37]. In a lognormal distribution, the expected (i.e., mean) value $\overline{X}$ of the lognormally distributed variable $X$ is:

$$\overline{X} = \exp\left(\mu + \frac{\sigma^2}{2}\right) \tag{10}$$

$\mu$ and $\sigma$ are the mean and standard variation of the natural logarithm of $X$.

The model parameters in Equation (9) depend on the scaling of the factors $x_i$. Normalizing the factors to the range $[-1;1]$ allows comparing the impact of the different main and interaction effects by simply comparing the corresponding $\alpha$ parameters. Table 5 lists the normalization conventions for the factors in the main programme. In this work, the superscript "$()^n$" indicates normalized factors, e.g., $R_t^n$ is the normalized $R_t$. For consistency, also the categorical factors like the environment $E$ are labelled similarly ($E^n$).

**Table 5.** Normalization of the factors in the main programme.

| Factor | Low Value (−1) | High Value (1) | Comment |
|---|---|---|---|
| $\varepsilon_r$ (%) | 0.6 | 1.2 | min. and max. values according to the test matrix |
| $\varepsilon_m$ (%) | 0 | 0.5 | min. and max. values according to the test matrix |
| $R_t$ (μm) | 0.194 | 65.5 | min. and max. values in the dataset |
| $t_h$ | no hold | incl.holds | categorical variable indicating if the test had holds (Table 2) |
| $E$ | air | PWR, VVER | categorical variable indicating the environment |

The aim of the current study is not only to obtain a numerical model that allows predicting the fatigue life of a specimen under a specific set of test conditions, but especially to determine which of the investigated factors have a significant impact on fatigue life. Therefore, the selected model should not only describe the data, but also include only those variables that have a significant effect on fatigue life. Many algorithms are available for fitting a model to the data. For the present study, we chose the backward elimination [38] method. This algorithm starts with a full model, including all factors and interactions that are being considered (in this case, a second order factorial model, Equation (9)) and evaluates the predictive performance of this model. In the next step, one model parameter (main effect or interaction) is removed, and the performance of the reduced model is evaluated. This procedure is repeated iteratively until only the intercept is left. This approach allows more easily comparing models with different numbers of factors than is the case for other algorithms that do not eliminate factors at all or where the number of factors is not changed in every step.

The model that best fits the data is not necessarily the most useful model since models with more parameters can easily overfit the data (i.e., fit the noise). Two approaches were used here for model selection. In the first approach, the data set is divided into a training set and a validation set. The data in the training set are used to determine the model parameters $\alpha_i$. The data in the validation set are then used to evaluate the predictive performance of the model. Since the data in the validation set were not used to determine the model parameters, the predictive performance of the model on the validation set is a good measure for the model performance under new conditions within the parameter range in which the model was optimized.

From Figure 2, it is clear that the data sets can be roughly separated into four distinct groups by the two levels of $\varepsilon_r$ and $E$. The training and validation sets are selected in such a way that 75% of the data in each of the four groups are in the training set and 25% in the validation set. This approach is shown in Figure 4a, where the -LogLikelihood for the training and the validation sets is plotted over the iteration steps of the algorithm. The -LogLikelihood, the negative natural logarithm of the likelihood function, is a measure for the goodness of fit, whereby smaller numbers indicate a better fit. The iteration steps of the algorithm start with Step Number 0, i.e., the full model including all main effects and all two parameter interactions. Moving on the abscissa left allows following the progression of the algorithm until at the leftmost step (here, Step 15), only the intercept remains.

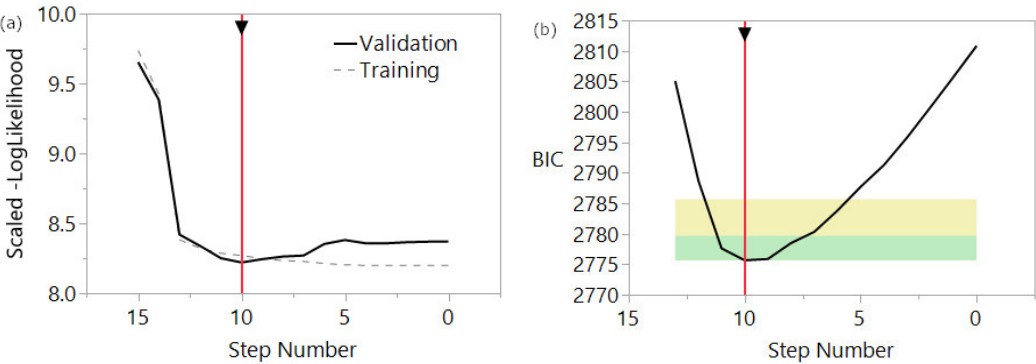

**Figure 4.** Comparison of the model performances as a function of the step in the algorithm, i.e., the number of factors that were removed from the model. Note that progression on the abscissa is from right to left. The vertical red line indicates the optimal model according to the algorithm. (**a**) -LogLikelihood for the training and validation sets. (**b**) BIC for the full data set; the green area indicates "very good" model performance (strong evidence that a model is comparable to the best model); the yellow area indicates "good" model performance (weak evidence that a model is comparable to the best model) [39].

The dashed line refers to the training set. The -LogLikelihood for the training set rises continuously with the progression of the algorithm (from right to left). This is expected since reducing the number of terms in the model will necessarily lead to worse fits. The behaviour of the solid curve for the validation set is different: initially, the -LogLikelihood drops until it reaches a minimum in Step 10 (indicated by the vertical red line) and continuously rises from there. This means that the model that best describes the validation set is reached in Step 10 of the algorithm. The corresponding model coefficients are listed in Table 6 Model (a) (Appendix A).

An alternative approach for selecting a model and to avoid overfitting is using a measure for the quality of the fit that penalises models with a larger number of parameters. The Bayesian information criterion (BIC) is such a measure. It is defined as:

$$\text{BIC} = -2\text{LogLikelihood} + k\ln(n) \tag{11}$$

where $k$ is the number of parameters in the model and $n$ the number of data points. As for the -LogLikelihood discussed above, lower values of BIC indicate a better fit. The second

term of the sum in Equation (11) penalizes models with more parameters. Figure 4b shows the BIC for the different steps in the backward elimination algorithm for the full data set. The best model is again reached in Step 10; the corresponding model coefficients are listed in Table 6 Model (b).

**Table 6.** Coefficients for the best models in Figure 4. Note that the normalized versions of the factors need to be used (Table 5); in the case of the categorical variable $E^n$ the coefficient is zero for $E^n = -1$ and the value in the table for $E^n = +1$. The $p$-value in the last column is an indication of the statistical significance of an effect; a threshold of 0.05 is often used as criterion for statistical significance with lower values indicating higher significance. $\sigma$ is a parameter in the lognormal distribution (Equation (10)).

| Model | Factor | Estimate | Std Error | $p$-Value |
|---|---|---|---|---|
| Model (a) | $I$ | 9.170 | 0.04524 | <0.0001 |
| | $\varepsilon_r^n$ | −0.9011 | 0.04644 | <0.0001 |
| | $E^n$ [+1] | −1.637 | 0.05123 | <0.0001 |
| | $R_t^n$ | −0.1995 | 0.04137 | <0.0001 |
| | $(\varepsilon_r^n - 0.06958) * E^n[+1]$ | 0.1444 | 0.05558 | 0.0094 |
| | $(\varepsilon_r^n - 0.06958) * (R_t^n + 0.51401)$ | 0.09537 | 0.04329 | 0.0276 |
| | $\sigma$ | 0.2850 | 0.02844 | <0.0001 |
| Model (b) | $I$ | 9.157 | 0.04059 | <0.0001 |
| | $\varepsilon_r^n$ | −0.9355 | 0.04578 | <0.0001 |
| | $E^n$ [+1] | −1.637 | 0.04594 | <0.0001 |
| | $R_t^n$ | −0.2169 | 0.03702 | <0.0001 |
| | $(\varepsilon_r^n - 0.06958) * E^n[+1]$ | 0.1766 | 0.05218 | 0.0007 |
| | $(\varepsilon_r^n - 0.06958) * (R_t^n + 0.51401)$ | 0.1097 | 0.04005 | 0.0062 |
| | $\sigma$ | 0.2913 | 0.02543 | <0.0001 |

### 3.1.4. Discussion

Comparing the model coefficients listed in Table 6 Models (a) and (b) shows that both models include the same terms, namely the main effects $\varepsilon_r^n$, $E^n$ and $R_t^n$, as well as the two interactions $\varepsilon_r^n * E^n$ and $\varepsilon_r^n * R_t^n$. The estimates for the coefficients of all three main effects are negative, indicating their detrimental effect on fatigue life.

The estimated factor for the interaction $\varepsilon_r^n * E^n$ is positive; large values for either $\varepsilon_r^n$ or $E^n$, i.e., large strain ranges or testing in the LWR environment therefore partly compensate the negative effects of $\varepsilon_r^n$ and $E^n$; at high strain ranges, there is less environmental effect. This is consistent with the observation reported in [3].

Similarly, the positive coefficient related to the interaction term $\varepsilon_r^n * R_t^n$ reduces the negative impact of a high surface roughness at high strain ranges. This is understandable: $R_t$ affects crack initiation rather than crack growth, so one would expect $R_t$ to have a more deleterious impact in situations where fatigue life is dominated by crack initiation, i.e., at low strain ranges, which is what the models predict.

Models (a) and (b) were determined using the same algorithm (backward elimination), but with different validation methods. Published and project internal analyses with different algorithms and slightly different data sets consistently showed the main effects $\varepsilon_r$, $E$ and $R_t$ to have the largest impact [40]. In most cases, one or two two-factor interactions were found to be statistically significant, but not practically relevant, i.e., they did not have a major impact on the predicted fatigue life. The interactions that were found to be statistically significant varied in evaluations with increasing size of the data set and depending on the algorithm used for the model optimization. This may indicate that the size of these effects is at the limit of what is detectable with the number of tests available in this work.

This is confirmed by the optimization curves (the solid black lines) for both models in Figure 4. In both cases, the best model is found in Step 10, but the performance of the models in Step 9 or 11 is very comparable. A further reduced model, including only the

main effects, was therefore calculated (using the BIC validation); the model parameters are listed in Table 7.

**Table 7.** Coefficients for a reduced model including only the main effects. Note that the normalized versions of the factors need to be used (Table 5); in the case of the categorical variable $E^n$ the coefficient is 0 for $E^n = -1$, and the value in the table for $E^n = +1$.

| Model | Factor | Estimate | Std Error | *p*-Value |
|---|---|---|---|---|
| Model (c) | $I$ | 9.173 | 0.04283 | <0.0001 |
| | $\varepsilon_r^n$ | −0.8354 | 0.02690 | <0.0001 |
| | $E^n$ [+1] | −1.650 | 0.05097 | <0.0001 |
| | $R_t^n$ | −0.2160 | 0.03696 | <0.0001 |
| | $\sigma$ | 0.3124 | 0.03103 | <0.0001 |

Figure 5a compares the $N_{25}$ predicted by the three models to the experimentally observed values. As could be expected from Table 6, Models (a) and (b) are hardly distinguishable. Only at very high $N_{25}$ do differences become apparent. Model (c), which only includes the main effects, differs visibly from the other two models. For high fatigue lives, Model (c) systematically predicts lower $N_{25}$, whereas the contrary can be observed in the medium $N_{25}$ range around 4000 cycles. In the region where $N_{25}$ is around 1000 cycles, all three models match well in general, with Model (c) deviating from the others in some cases. These differences result from omitting the interaction effects. However, the differences between the reduced model (c) and the optimal models (a) and (b) is small compared to the scatter observed experimentally. Therefore, Model (c) seems to be good enough to make realistic predictions.

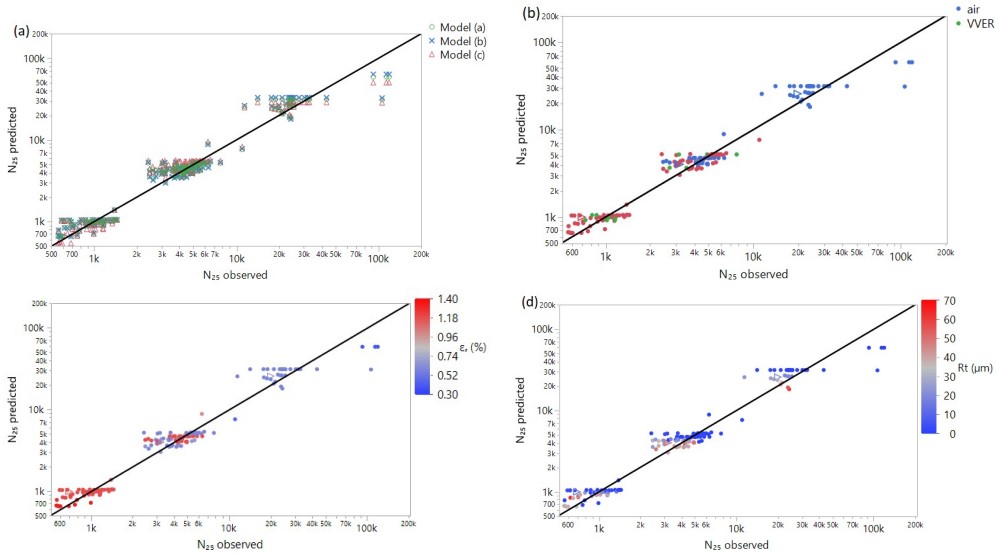

**Figure 5.** Model predictions for $N_{25}$ vs. experimental values: (**a**) comparison between the models (**a**–**c**). For the predictions with Model (**a**), colour coding highlights the different environments (**b**), strain ranges (**c**) and the surface roughnesses (**d**).

During the analysis, all tests were considered to be either carried out in air or in the LWR environment, where the LWR environment included simulated PWR, as well as simulated VVER conditions, and no distinction was made between the latter two. Furthermore, all tests in the VVER environment (and only these) were performed on a 321 steel. The question is if considering the PWR and VVER tests was a sensible approach. Figure 5b–d compares the predicted $N_{25}$ from Model (a) to the experimentally observed values, whereby the colour coding indicates the different environments, strain ranges and surface roughnesses.

The VVER data in Figure 5b are distributed around the black reference line and do not show any particularities. Hence, based on the data available here, the model describes the VVER data just as well as the PWR data. Similarly, the model predictions work equally well for different strain ranges $\varepsilon_r$ (c) and surface roughnesses $R_t$ (d). The effect of $R_t$ on the predicted fatigue life is visible by the separation of the blue points with very low and the grey/red points with higher $R_t$ values. The gap between these two groups is higher for larger fatigue lives, showing the interaction between $R_t$ and $\varepsilon_r$.

### 3.2. Sub-Programme on Low $F_{en}$ Conditions

#### 3.2.1. Test Conditions

In this sub-programme, a limited number of tests were carried out at conditions with a lower $F_{en}$ than in the main programme. From Equation (3), it follows that without changing the water chemistry (i.e., the DO content), the approaches that allow reducing $F_{en}$ are reducing the temperature $T$ and increasing the (positive) strain rate $\dot{\varepsilon}$. The maximum strain rate that could be achieved in all autoclaves in the project was an increase by a factor 10 compared to the main programme, i.e., $\dot{\varepsilon} = 0.1\%/s$. This leads to a $F_{en} = 2.68$; the same $F_{en}$ is obtained by reducing $T$ to 230 °C (Table 2).

#### 3.2.2. Data Overview

Only a limited number of tests was available for the test programme at reduced $F_{en}$. Here, only tests with a strain range $\varepsilon_r = 0.6\%$ in the LWR environment are considered. Forty-nine tests were available for the analysis [41], of which 15 were at the lower $F_{en}$, with eight tests at reduced temperature $T$ and seven tests at increased positive strain rate $\dot{\varepsilon}$. Data from the main programme at the positive strain rate 0.01 %/s were used as reference data. Some of these tests were carried out with mean strain or hold times. However, since the analysis of the data in the main programme did not reveal any mean strain or hold time effects, these parameters are not considered in context with the low $F_{en}$ data. The fatigue lives of the tests used in the low $F_{en}$ analysis are plotted in Figure 6 and the distributions of the most relevant test parameters in Figure 7.

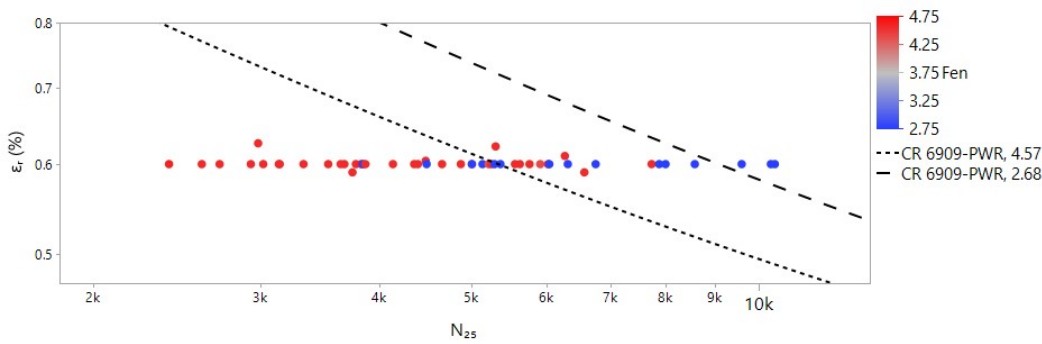

**Figure 6.** Data in the low $F_{en}$ programme; the reference curves are calculated from the NUREG/CR-6909 mean air curve and the two $F_{en}$ values considered here.

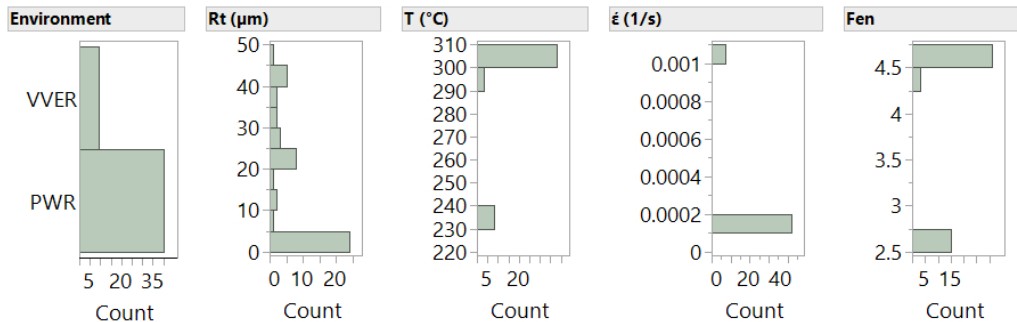

**Figure 7.** Distribution of the factors in the low $F_{en}$ programme.

### 3.2.3. Data Analysis

Because of the limited number of tests available for this sub-programme, no tests with reduced temperature $T$ and increased strain rate $\dot{\varepsilon}$ were carried out. This gap in the test matrix is reflected in the correlation between $T$ and $\dot{\varepsilon}$ in the correlation matrix (Table 8). It should also be noted that the small number of tests led to a reduced spectrum of $R_t$ in the low $F_{en}$ data. For both groups of low $F_{en}$ data (with reduced $T$ and with increased $\dot{\varepsilon}$), the maximum $R_t$ is around 30 μm, whereas for the group at higher $F_{en}$, it is almost 50 μm.

**Table 8.** Correlations between the factors in the low $F_{en}$ programme

|  | $R_t$ | $\dot{\varepsilon}$ | $T$ |
|---|---|---|---|
| $R_t$ | 1.0000 | −0.0589 | 0.0823 |
| $\dot{\varepsilon}$ | −0.0589 | 1.0000 | 0.1846 |
| $T$ | 0.0823 | 0.1846 | 1.0000 |

Table 9 shows the normalization definitions for the factors in the low $F_{en}$ programme. Notice that the spectrum of $R_t$ values is smaller than in the main programme, which leads to a slightly different normalization.

**Table 9.** Normalization of the factors $T$ and $\dot{\varepsilon}$ for the tests at reduced $F_{en}$.

| Factor | Low Value (−1) | High Value (1) | Comment |
|---|---|---|---|
| $T$ (°C) | 230 | 302.3 | min. and max. values in the dataset |
| $\dot{\varepsilon}$ (%/s) | 0.01 | 0.1 | min. and max. values according to test matrix |
| $R_t$ (μm) | 0.335 | 49.75 | min. and max. values in the dataset |

The small number of tests available in the two low $F_{en}$ groups makes it impractical to divide the data into a training and a validation set. Therefore, only the BIC method described in Section 3.1.3 is used for the analysis of the low $F_{en}$ data. As before, the optimal model is determined by means of the backward elimination algorithm. The initial model includes all three main effects $R_t^n$, $\dot{\varepsilon}^n$ and $T^n$, as well as the two-factor interactions $R_t^n*\dot{\varepsilon}$ and $R_t^n*T^n$. Since no data with high $\dot{\varepsilon}$ and low $T$ are available, no information about a possible interaction between these two parameters is present in the data.

The plot with the different steps of the backward elimination algorithm is shown in Figure 8; the parameter estimates for the optimal model, which includes only the main effects, are listed in Table 10.

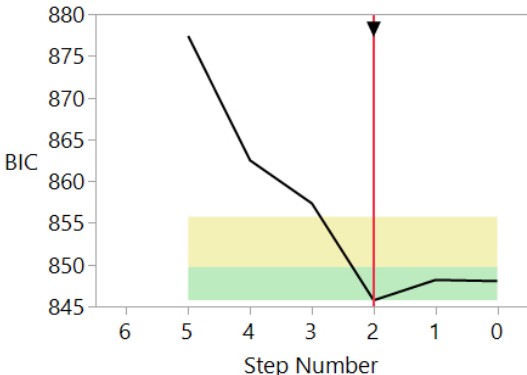

**Figure 8.** Variation of BIC during the iteration steps of the backward elimination algorithm for the low $F_{en}$ model.

**Table 10.** Coefficients of the optimal model for the low $F_{en}$ data. Note that the normalized versions of the factors need to be used (Table 9).

| Factor | Estimate | Std Error | *p*-Value |
|--------|----------|-----------|-----------|
| $I$ | 8.643 | 0.05659 | <0.0001 |
| $R_t^n$ | −0.2879 | 0.05369 | <0.0001 |
| $\dot{\varepsilon}^n$ | 0.2048 | 0.05023 | <0.0001 |
| $T^n$ | −0.2091 | 0.03572 | <0.0001 |
| $\sigma$ | 0.2303 | 0.02785 | <0.0001 |

### 3.2.4. Discussion

As would be expected, higher $\dot{\varepsilon}$, as well as lower $R_t$ and $T$ increased the fatigue life. From the coefficients in Table 10, it is clear that increasing $\dot{\varepsilon}$ from 0.0001/s to 0.001/s and reducing $T$ from 300 °C to 230 °C had the same beneficial effect on fatigue life. In the range of parameters studied here, the effect of $R_t$ was ca. 40% stronger than that of the other two parameters.

In Table 11, the fatigue lives for polished specimens ($R_t^n = -1$) are calculated for different settings of $T^n$ and $\dot{\varepsilon}^n$. The first row corresponds to the conditions in the high $F_{en}$ programme. In the second and third row, the fatigue lives for the two means of reducing the $F_{en}$ are calculated. As expected, reducing $T^n$ and increasing $\dot{\varepsilon}^n$ yield virtually the same predicted $N_{25}$. The ratio between the predicted $N_{25}$ values at low and high $F_{en}$ conditions is 1.5. This is reasonably close to the ratio of the high and low $F_{en}$ values (1.7).

**Table 11.** Fatigue lives calculated with the model in Table 10.

| $R_t^n$ | $\dot{\varepsilon}^n$ | $T^n$ | $N_{25}$ |
|---------|------------------------|-------|----------|
| −1 | −1 | 1 | 5131 |
| −1 | −1 | −1 | 7794 |
| −1 | 1 | 1 | 7728 |

Other algorithms led to models that also included the statistically significant interactions $R_t^n{*}\dot{\varepsilon}$ and $R_t^n{*}T^n$. In these models, the coefficient for $R_t^n{*}\dot{\varepsilon}$ was negative, and the coefficient for $R_t^n{*}T^n$ was positive. That would mean that in the parameter range studied here, the fatigue life of polished specimens would be more sensitive to the variations of positive strain rate and temperature than the fatigue life of specimens with a ground surface finish. However, the error bars of the more complex models overlap with the error bars of the simpler model in Table 10 so that from an application point of view, there is no practical difference between the models, and the simpler one can be used.

### 3.3. Sub-Programme on Hold Times

A series of tests that included hold times was performed in Phases I and II of the INCEFA-PLUS test programme (Section 3.1). The tests on hold time effects in Phases I and II were carried out in strain-control at the strain ranges 0.6% and 1.2% (Table 2). The hold periods were introduced at the position in the cycles where the mean strain (0% or 0.5%) was reached with a positive strain rate. Holds of 72 h were introduced in three cycles per tests; the cycles with holds depended on the test conditions. For tests in air, holds were added at 6000 cycle intervals starting from the 6000th cycle for strain range 0.3% and 1000 cycle intervals starting from the 1000th cycle for 0.6%. Preliminary analysis including Phase II tests ([40,42], confirmed in Section 3.1.3) suggested that there was no observable effect of hold times for the tested conditions, although beneficial effects of hold time on fatigue life have been reported by the AdFaM (Advanced Fatigue Methodologies) project [13].

Therefore, hold times were removed from Phase III testing, and in parallel, a sub-programme on hold time effects was initiated where the test conditions were more closely aligned to those used in the AdFaM study [13].

#### 3.3.1. Data from Hold Time Testing

According to [13], hold time effects were most prominent at low strain amplitude and with holds at zero stress at elevated temperature. To increase the chances of observing a hold time effect, the strain range in the sub-programme on hold time effects therefore was reduced to 0.4%. Furthermore, holds were performed under zero load rather than in strain control (in some cases, at non-zero mean strain) as in the main program. The holds consisted of three 72 h holds at 350 °C at 10,000 cycle intervals starting from the 10,000th cycle. The temperature during hold times was increased from 300 °C to 350 °C. Cycling was carried out at room temperature or at 300 °C. All tests were performed in air on the common batch XY182 of 304L [43].

Figure 9 shows the evolution of the maximum stress per cycle during the test.

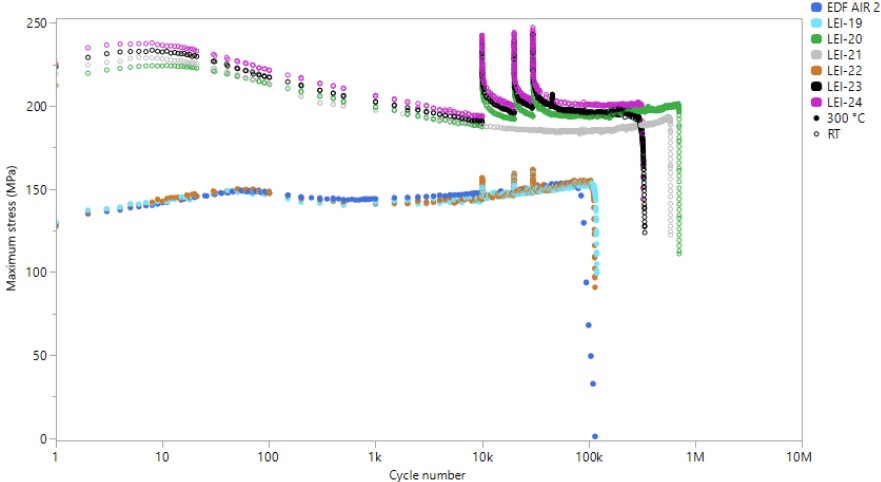

**Figure 9.** Maximum stress as a function of cycle; all tests were carried out at a strain range of 0.4%. The tests "EDF AIR 2" and "LEI-21" are the only tests that did not have holds.

Three of the seven tests (specimens "EDF AIR 2", "LEI-19" and "LEI-22") were carried out at 300 °C, whereas the other four were performed at room temperature. The tests with the specimens "EDF AIR 2" and "LEI-21" were the only ones without holds.

#### 3.3.2. Discussion of Hold Time Data

The reducing effect of the increased temperature on the stress level and fatigue life is obvious from Figure 9. The hardening effect of the hold periods is visible from the

peaks in the maximum stress at 10,000, 20,000 and 30,000 cycles. The curves for the tests at 300 °C showed a primary hardening followed by softening and secondary hardening before failure occurred around Cycle 100000. The maximum cyclic stresses of the three tests evolved in very similar manner—especially given that they were tested in two different laboratories (LEI and EDF). The hold times led to hardening, but there was no long lasting effect in either stress level or fatigue life.

The situation for the tests at room temperature was different: until the first hold at Cycle 10000, the stress curves evolved in parallel even if there were some differences in absolute stress values. The first hold (at 350 °C at zero stress) then hardened the material, similar to what was observed for the tests cycled at 300 °C. However, the stress increase was much higher and decayed more slowly when cycling restarted. Furthermore, for the remainder of the tests, the three tests with holds reached higher stress levels compared to the reference test ("LEI-21") than they had before the holds. The second and third hold times seemed to have less effect. The increased stress, however, did not seem to have an impact on fatigue life. In particular, no extension of fatigue life as reported in [13] was evident (two of the three tests with holds were actually shorter than the reference test without holds). The reason for that discrepancy with the AdFaM results remains unclear; it might be that the number of hold periods played a role. In the tests reported in [13], hold periods were applied throughout the test, so depending on the conditions, there were many more than just three hold periods in a test.

## 4. Conclusions

A major test programme on strain controlled fatigue in air and LWR conditions was carried out. The main programme with $F_{en}$= 4.57 investigated the effects of strain range, mean strain, hold time, surface roughness and environment on the fatigue life of austenitic steels. The test matrix was optimized by the design of experiments methodology. A linear model taking into account possible interactions was determined. No influences of hold time and mean strain were identified. The test data could be described by a model including only the main factors strain rate, environment and surface roughness. The interaction effects of strain range with the environment, as well as surface roughness were found to be statistically significant, but of limited practical relevance.

In a sub-programme at a lower $F_{en}$ = 2.68, the influences of temperature, positive strain rate, as well as surface roughness were studied. Because of the limited number of tests, not all possible interactions could be addressed. No firm evidence for an interaction of surface roughness with either temperature or strain rate was detected. In the parameter range investigated, the effect of surface roughness was slightly larger than the effects of temperature and strain rate. As predicted by NUREG CR-6909, the reduction of the temperature from 300 °C to 230 °C in the LWR environment was found to have the same effect on fatigue life compared to $F_{en}$ = 4.57 as the increase of the positive strain rate from 0.0001/s to 0.001/s.

Finally, a limited number of tests in air with holds at elevated temperature under no stress conditions did not find evidence for beneficial effects of hold times on fatigue life like those found in another study. The reason might be the difference in the number of hold time periods.

**Author Contributions:** Conceptualization: M.B. methodology: M.B. validation: A.M. formal analysis: M.B., J.M. and A.M. investigation: S.A., R.C., W.-J.C., L.D., G.D., M.E., A.G., C.H., A.M., R.N., F.J.P., N.P., J.-C.l.R., P.S. and M.V. data curation: T.A., M.B., R.C., L.D., C.H., A.M., N.P., J.-C.l.R., P.S., C.T.C., M.T. and M.V. writing—original draft preparation: M.B. and G.D. writing—review and editing: W.-J.C., M.V., A.M., K.M., C.H., G.D. and P.S. project administration: K.M., M.B., R.C., C.H., J.-C.l.R. and M.V. funding acquisition: K.M. and R.C. All authors read and agreed to the published version of the manuscript.

**Funding:** This project received funding from the Euratom Research and Training Programme 2014-2018 under Grant Agreement No. 662320.

**Institutional Review Board Statement:** Not applicable.

**Informed Consent Statement:** Not applicable.

**Data Availability Statement:** Requests to access the data presented in this study can be submitted to the data owner(s). The data are stored in a database and traceable through DOIs [32,41,43] but are not publicly available due to the data policy of the project.

**Conflicts of Interest:** The authors declare no conflict of interest. The funders had no role in the design of the study; in the collection, analyses, or interpretation of data; in the writing of the manuscript; nor in the decision to publish the results.

## Abbreviations

The following abbreviations and symbols are used in this manuscript:

| | |
|---|---|
| AdFaM | Advanced Fatigue Methodologies (name of a project) |
| DOE | design of experiments |
| EAF | environmentally assisted fatigue |
| LTO | long-term operation |
| NPP | nuclear power plant |
| PWR | pressurized water reactor |
| VVER | voda-vodyanoi energetichesky reaktor (Russian: pressurized water reactor) |
| $\alpha_i$ | model parameter |
| BIC | Bayesian information criterion |
| DH2 | dissolved hydrogen content |
| DO | dissolved oxygen content |
| $E$ | categorical variable for environment; either "air" or "LWR" |
| $E^n$ | normalized categorical variable for environment; either "$-1$" or "$+1$" |
| $\varepsilon_r$ | strain range: difference between the maximum and minimum strain during a test |
| $\varepsilon_r^n$ | normalized strain range |
| $\dot{\varepsilon}$ | strain rate |
| $\dot{\varepsilon}^n$ | normalized strain rate |
| $\dot{\varepsilon}^*$ | strain rate parameter defined in CR-6909 [4] |
| $\varepsilon_m$ | mean strain: strain level in the middle between the maximum and minimum strain in a strain controlled test |
| $\varepsilon_m^n$ | normalized mean strain |
| $F_{en}$ | environmental factor |
| $I$ | intercept in model |
| LogLikelihood | natural log of the likelihood function |
| $N_{25}$ | fatigue life, 25% force drop compared to stabilized linear behaviour |
| $N_f$ | fatigue life |
| $N_{f,air,RT}$ | fatigue life in air at room temperature |
| $N_{f,LWR}$ | fatigue life in LWR environment |
| $N_X$ | fatigue life, X% force drop |
| $O^*$ | dissolved oxygen parameter defined in CR-6909 [4] |
| $r_{i,j}$ | correlation between two variables $x_i$ and $x_j$ |
| $R_a$ | average surface roughness as defined in ISO 4287 [30] |
| $R_t$ | maximum roughness height as defined in ISO 4287 [30] |
| $R_t^n$ | normalized surface roughness Rt |
| $\sigma_e$ | electric conductivity |
| $t_h$ | categorical variable for hold time |
| $t_h^n$ | normalized hold time |
| $T$ | temperature |
| $T^n$ | normalized temperature |
| $T^*$ | temperature parameter defined in CR-6909 [4] |

## Appendix A

---

**Algorithms A1** Matlab Function for Model (a)

---

```
function N25 = model_a(epsn,Rtn,En)
% MODEL_A
% This function calculates the fatigue life according to the model (a) in Table 6
% The entries are the normalized values of strain range, surface
% roughness and environment as in Table 5.

% The coefficients according to Table 6 (a)
% main effects and sigma
I = 9.1695785;
epsn_coeff = −0.90106;
switch En
    case −1
        En_coeff = 0;
    case 1
        En_coeff = −1.637142;
end
Rtn_coeff = −0.199527;
sigma = 0.2849506;

% coefficients and offsets for the interaction terms
% interaction between epsn and En
switch En
    case −1
        epsn_x_En_coeff = 0;
    case 1
        epsn_x_En_coeff = 0.144387;
end
epsn_offset1 = −0.06958;
% interaction between epsn and Rtn
epsn_x_Rtn_coeff = 0.0953654;
epsn_offset2 = −0.06958;
Rtn_offset = 0.51401;

N25 = exp(I + epsn_coeff*epsn + En_coeff*En + Rtn_coeff*Rtn +...
    epsn_x_En_coeff*(epsn + epsn_offset1)*En +...
    epsn_x_Rtn_coeff*(epsn + epsn_offset2)*(Rtn + Rtn_offset) +...
    sigma^2/2);
end
```

---

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
