# Peer review of "Characterization of Austenitic Stainless Steels with Regard to Environmentally Assisted Fatigue in Simulated Light Water Reactor Conditions"

_metals, doi:10.3390/met11020307_

Round 1
Reviewer 1 Report
I highly recommend publishing this excellent paper. It is of a very good quality. The originality is clear and reference to similar studies are given. The outstanding amount of test data, the good quality of analyzing the data and summarizing the results via simple factorial models are of great significance. Public interest is assured, because of possible consequences in extending operation times for nuclear power plants.
I have some minor remarks. The authors should consider the following points.
Pate 1 Line 9-10: Statistical evidence for the impact of the interactions of… with … as well as…
The sentence is not easily understandable without reading the paper. Maybe writing two sentences can clarify.
Page 2 line 43: “…in a sub-study at reduced Fen”
At this point I didn’t understand the sentence. Fen is introduced as an outcome of the tests (Formula 1). So one cannot “set” Fen (as presented later in Formula 2). So here it would be helpful to write something like “In a sub-study with different testing parameters resulting in a reduced Fen… (see Formula 2)…”
Page 2 Line 47: “…affected by this difference”.
I didn’t understand what is meant here by “difference”. Smaller dataset? Earlier (different?) data evaluation? Different outcome?
Page 3 Table 1: Cr and Ni-contents for 304L.
Probably the measurement accuracy is better than whole numbers for Cr and Ni, so one could write e.g. Cr 18.0 or 18.00 if this is approx. the measurement accuracy. Please check also other numbers.
Page 3 line 55/56 and line 72/73: These sentences are repetitions from introduction.
It is probably OK to repeat here, but please reread 1-Introduction and 3-Test programme. Maybe some sentences in Introduction can be shortened or deleted.
Page 4 Table 3: Maybe it helps to move Table 2 to 3.1.1 for better understanding (Or refer to Table 2 in 3. Test programme in the text.)
Page 4 Line 103: “the surface roughness characteristics maximum roughness…”
Is correct but maybe using different sentence structure might help understanding.
Page 4 Line 109: VVER
The abbreviation is not introduced in the text.
Page 5 Table 3: “Variations …may occur”
If known, parameters and the range of these variations should be mentioned.
Page 10 Table 6: Caption
In the caption it should be mentioned that we see the coefficients of “model (a) and model (b)”, as these are referenced in the text.
Page 10 Table 7: Caption
In the caption it should be mentioned that coefficients of “model (c)” are given
Page 11 Figure 5: (a) Points for model (a)
It is impossible to see the points for model (a). Please consider partial transparency for points or smaller points or further options (further plots).
Page 15 Figure 9: Please choose different colors for either LEI23 or LEI20 (e.g. grey or orange).
Page 15 Line 325: “…different laboritories”
The different labs should be mentioned somewhere in the text. One can only guess that LEI and EDF are different labs from captions in Fig 9?
Page 16 Line 357: Please check if “under no stress” is correct here. As I understood, hold times were at mean strain.
Reviewer 2 Report
Comments and Suggestions for Authors
This study evaluates the environmentally assisted fatigue behavior of various austenitic stainless steels in light water reactor condition. The topic of this study is interesting, however, drawbacks are found throughout the article. Firstly, the literature review is poor and superficial. In the Introduction section, the background information regarding to the environmentally assisted fatigue of austenitic stainless steels is very limited. In this manuscript, it seems that the authors focus more on their previous work rather than reviewing other people’s work. A brief and comprehensive literature review related to this study is needed to help the readers better understand the aim of this study. Secondly, there are 17 references in this manuscript cite from the co-authors’ previous work. Please note that the inappropriate self-citations could lead to serious ethical problem. Thirdly, the experimental procedure, especially for the fatigue test, is not mentioned in the manuscript. The geometric dimension of the testing specimen, the preparation of the sample and the equipment used for the fatigue test are completely missing in the manuscript. In addition, testing parameters such as “strain range (εr)”, “mean strain (εm)” and “hold time (th)” and the method for data analysis are not clearly presented in the manuscript. The authors should give a detail description on the experimental procedure, the testing parameter as well as the data analysis in the manuscript for the readers’ reference. Fourthly, there is a lot of unnecessary statement describing project-related matter. For example,
Page 3: “All test data were uploaded in a dedicated materials database operated by the European Commission (MatDB) and are accessible to all project partners.”
Page 4: “As an additional quality assurance measure, each test was validated by a panel of fatigue experts from within the project and rated with regard to the quality of the test and the completeness of the information in the database. The test quality was determined on the basis of data like the cyclic stress amplitudes and hysteresis curves [23]. Where necessary, this information was complemented by microstructural characterization of the specimens [24]. From the strain controlled tests carried out during the project 94% received a quality rating of 1 or 2 (out of 4) and were accepted without restriction for analysis.”
Other examples such as Figure 3 and Figure 6, which do not contain any important results related to the topic of this study. These unnecessary information do not actually help the readers better understand the detail of this work; on the contrary, they make this manuscript too long and hard to read. The authors should focus more on the study itself, rather than putting too much effort describing the project-related matter. Lastly, the model presented in this manuscript is questionable, because there are only a few testing conditions for each factor, and the number of the test is limited. I strongly suggest the authors carefully examine and confirm the validity of the model before practical application, because for the nuclear power plant, any evaluation should be very conservative. Moreover, the manuscript contain many grammatical error, unclear statement and incorrect punctuation, which also make the readers difficult to understand. I strongly recommend the manuscript polished by a native English speaker to make it more readable. Based on the above comments, I do not recommend publication of this manuscript. Some of my other comments are listed below:
Page 2: “However, low cycle fatigue lives predicted by CR-6909 do not reflect current pressurized water reactor (PWR) plant experience where no failures attributed to environmental fatigue have been observed so far where the loading conditions were known [1].”
If there are no failures in the nuclear power plants attributed to environmental fatigue, the environmentally assisted fatigue would not be an issue. The authors should briefly describe why they studying the environmentally assisted fatigue of the austenitic stainless steels in the light water reactor condition for the readers’ reference.
Page 2: “Accordingly, EAF has received much attention in the last years [5–12]. Work by Chopra et al. presents a recent review with focus on ASME Code section III [2].”
Please briefly describe the important findings of the previous studies in the Introduction.
Page 3: “The large majority (86%) of the tests were carried out on a single batch (XY182 sheet 23201) of 304L stainless steel produced by Creusot Loire Industries. The remaining tests were carried out on a single batch of 321 (8%) and different batches of 304, 304L and 316L.”
There are 14% of the tests used different batches or materials. Could this affect the validity of the model?
Page 3: “All materials have been annealed at temperatures between 1050 °C and 1100 °C.”
Please indicate the annealing time in the sentence as well.
Page 3: “These parameters were selected based on on the interest of the project partners and the EPRI gap report [1].”
Please revise this sentence to “These parameters were selected based on the interest of the project partners and the EPRI gap report [1].”
Page 3: “However because of shake down early during the test, the mean strain does not have a significant effect on fatigue life.”
Please revise this sentence to “However, because of shake down early during the test, the mean strain does not have a significant effect on fatigue life.”
Page 3: “To investigate whether differences in the application of holds led to these differences a limited sub-programme on hold time tests was started (section 3.3).”
Please revise this sentence to “To investigate whether differences in the application of holds led to these differences, a limited sub-programme on hold time tests was started (section 3.3).”
Page 3: “In the absence of a dedicated standard for EAF tests in LWR conditions the tests were performed as far as possible according to…”
Please revise this sentence to “In the absence of a dedicated standard for EAF tests in LWR conditions, the tests were performed as far as possible according to…”
Page 3: “All test data were uploaded in a dedicated materials database operated by the European Commission (MatDB) and are accessible to all project partners.”
This is an unnecessary sentence and therefore it should be deleted.
Page 4: “As an additional quality assurance measure, each test was validated by a panel of fatigue experts from within the project and rated with regard to the quality of the test and the completeness of the information in the database. The test quality was determined on the basis of data like the cyclic stress amplitudes and hysteresis curves [23]. Where necessary, this information was complemented by microstructural characterization of the specimens [24]. From the strain controlled tests carried out during the project 94% received a quality rating of 1 or 2 (out of 4) and were accepted without restriction for analysis.”
There is no need to indicate the quality assurance in the manuscript. Please delete this paragraph.
Page 4: “With the exception of surface roughness, for which that was not possible technically, all factors are tested on two levels.”
Please explain why surface roughness is an exception in this experiment?
Page 4: “For all specimens the surface roughness characteristics maximum roughness height Rt [27] and average roughness Ra [27] values are available. While for polished specimens in most cases generic roughness values were used, all ground specimens were measured individually.”
The meaning of this sentence is unclear. Please revise. Furthermore, the authors do not describe how the surface roughness was measured and how specimens with different surface roughness were prepare.
Page 4: “In this work Rt is used rather than Ra because that is the parameter that can be expected to have more impact on crack intiation: a deeper scratch leads to larger stress concentration which facilitates crack intitation.”
Please define the surface roughness parameter “Rt” and “Rt” in the manuscript. In addition, please revise “intiation” and “intitation” to “initiation”.
Page 4: “As both values are strongly correlated (Figure 1) the choice of surface characteristic is not expected to have a major impact on the analysis.”
Please revise this sentence to “As both values are strongly correlated (Figure 1), the choice of surface characteristic is not expected to have a major impact on the analysis.”
Page 5: “According to CR-6909 the Fen for austenitic stainless steels can be formulated as [3]:”
Please revise this sentence to “According to CR-6909, the Fen for austenitic stainless steels can be formulated as [3]:”
Page 5: “For the test conditions in the main programme (Tables 2 and 3), Equation 2 yields Fen = 4.57.”
What is the value of “T” and “ε” used for the calculation?
Page 6: “While the majority of the tests in the environment were carried out using solid specimens in autoclaves, some data were acquired on hollow specimens where the water flows through the specimen. For hollow specimens, Nf is generally the cycle where leakage occurs. This is considered a rough equivalent to N25 [3]. There is an ongoing discussion whether the fatigue lifes from hollow and solid specimens can be compared directly [29–31]. A study carried out within INCEFA-PLUS led to the conclusion that no significant effect on the mean values is expected for the data discussed here [32]. (This analysis was done on an earlier (smaller) data set, but has been confirmed with the final dataset.) Therefore no further distinction between the two types of specimens is made here. For hollow specimens the strains are used directly as measured and no strain correction as suggested in [31] was applied.”
What is the exact geometry of the fatigue specimen used in this study? I do not understand why the authors carry out the comparison on the fatigue life of the hollow and solid specimen. This entire paragraph seems redundant to me.
Page 7: “Before starting the actual data analysis, it is useful to check for possible correlations between the input parameters.”
How did you check the correlations between the input parameters?
Page 7: “Strong correlations between the inputs can easily lead to wrong conclusions during the evaluation.”
Please explain why.
Page 7: “The three phases of the main test programme have been optimized by the design of experiments method [26] which also minimises the correlation between the factors.”
Please explain why.
Page 7: “Table 4 lists the correlations between the factors in the main programme.”
Please describe how you calculated the correlations between the factors.
Page 7: “An anticorrelation between εm and Rt is expected since in phases II and III no tests with holds were carried out anymore whereas in phase II a higher surface roughness Rt was introduced. Therefore, one would expect tests with holds to have on average lower Rt and hence an anticorrelation between Rt and th.”
I do not understand what the authors try to present here. Please revise this sentence to make it more readable.
Page 7: “The correlation between εr and Rt, however, is unexepected. Most likely it is a random effect resulting from the grinding process that was used to produce the rough surface finishes, and which yields a distribution of surface roughnesses rather than specific Rt values (Figure 3).”
Why surface roughness could affect the correlation between εr and Rt? In addition, please clearly describe the grinding process used in this experiment.
Page 7: “The xi are the different factors (such as Rt). The parameters αi and αij are the model parameters for the main effects and the two factor interactions and I is the intercept. For every test an equation like Equation 7 is formulated.”
This model is questionable because there are only a few testing conditions for each factor. The validity of the results should be carefully examined and confirmed.
Page 8: “For consistency also the categorical factors like the environment E are labelled similarly (En).”
Do you mean “For consistency, the categorical factors like the environment E are also labelled similarly (En).”?
Page 8: “For the present study we have chosen the Backward Elimination [34] method. It starts with a full model (in this case a second order factorial model, Equation 7) and with every iteration step removes one factor from the model until only the intercept is left.”
The authors should give more information about the “Backward Elimination [34] method” for the readers’ reference.
Page 8: “Two approaches have been used here for model selection. In the first approach, the data set is divided into a training set and a validation set. The data in the training set is used to determine the model parameters αi whereas the predictive performance of the model for the data in the validation set is used for model selection.”
More detail regarding to the first approach should be indicated in the manuscript for the readers’ reference.
Page 10: “In most cases one or two two-factor interactions have been found to be statistically significant but not practically relevant…”
Please revise this sentence to “In most cases, one or two two-factor interactions have been found to be statistically significant but not practically relevant…”
Page 11: “During the analysis all tests were considered to be either carried out in air or in LWR environment,…”
Please revise this sentence to “During the analysis, all tests were considered to be either carried out in air or in LWR environment,…”
Page 11: “In this sub-programme a limited number of tests were carried out at conditions with a lower Fen than in the main programme. From Equation 2 it follows that without changing the water chemistry (i.e. the DO content) the approaches that allow reducing Fen are reducing the temperature T and increasing the (positive) strain rate ε.”
Please revise this sentence to “In this sub-programme, a limited number of tests were carried out at conditions with a lower Fen than in the main programme. From Equation 2, it follows that without changing the water chemistry (i.e. the DO content), the approaches that allow reducing Fen are reducing the temperature T and increasing the (positive) strain rate ε.”
Page 13: “From the coefficients in Table 10 it is clear that,…”
Please revise this sentence to “From the coefficients in Table 10, it is clear that…”
Page 14: “Preliminary analysis including phase II tests ([36,38], confirmed in section 3.1.3) suggested that there was no observable effect of hold times for the tested conditions although beneficial effects of hold time on fatigue life have been reported from the AdFaM project [17].”
The abbreviation “AdFaM” should be spelled out upon first appearance in the manuscript.
Page 14: “From the 7 tests, 3 (EDF AIR 2, LEI-19 and LEI-22) were carried out at 300 °C whereas the other 4 were performed at room temperature.”
The abbreviation “AIR” and“LEI” should be spelled out upon first appearance in the manuscript.
Table 1:
The unit of the chemical composition is missing (wt.% or at.%).
Table 2:
Please explain in the manuscript how these testing conditions are chosen in this study. In addition, testing parameters such as “strain range (εr)”, “mean strain (εm)” and “hold time (th)” are not defined in the manuscript. Please define all these testing parameters.
Table 3:
Please define “DH2” and “DO” that shown in the table. I am not quit understand why the authors state that “Variations because of national interests may occur; these are listed in the database.” Please explain this in the manuscript.
Table 6:
Please explain how the “p-value” is calculated. Does higher p-value indicate higher significance of the effect?
Figure 3:
In this figure, I do not see any important results related to this study. Therefore, this figure should be deleted.
Figure 5:
The authors should present the results of the εr and Rt as well since there are both important factors that could affect the fatigue life.
Figure 6:
The color bar should be deleted since there is only two Fen in this figure, 2.68 and 4.57.
Figure 7:
This figure should be deleted because it does not contain any important results related to this study.
Reviewer 3 Report
The presented test results and the analysis of these results may constitute an important source of data needed to assess the fatigue life of the considered materials under operating conditions, while ensuring the possibility of making decisions regarding the further operation of the existing devices, taking into account the possible dispersion of material properties due to differences in the chemical composition and technology of their production.
Round 2
Reviewer 2 Report
Comments and Suggestions for Authors
Good job. I am surprised that the authors respond all the comments in such a short period. After careful examination of the revised manuscript, I have no further question about this work. The contents of the current manuscript has met the requirement of the journal and worth for publication. Congratulations and keep on the good work.